# The social specificities of hostility toward vaccination against Covid-19 in France

**Nathalie Bajos** *[ID]*[⊕], **Alexis Spire**[ID][⊕], **Léna Silberzan**[ID], **for the EPICOV study group**[¶]

IRIS, Inserm/EHESS/CNRS, Aubervilliers, France

⊕ These authors contributed equally to this work.
¶ Membership of the EPICOV study group is provided in the Acknowledgments.
* nathalie.bajos@inserm.fr

## Abstract

Equal Access to the COVID-19 vaccine for all remains a major public health issue. The current study compared the prevalence of vaccination reluctance in general and COVID-19 vaccine hesitancy and social and health factors associated with intentions to receive the vaccine. A random socio-epidemiological population-based survey was conducted in France in November 2020, in which 85,855 adults participants were included in this study. We used logistic regressions to study being "not at all in favor" to vaccination in general, and being "certainly not" willing to get vaccinated against Covid-19. Our analysis highlighted a gendered reluctance toward vaccination in general but even more so regarding vaccination against COVID-19 (OR = 1.88 (95% CI: 1.79–1.97)). We also found that people at the bottom of the social hierarchy, in terms of level of education, financial resources, were more likely to refuse the COVID-19 vaccine (from OR = 1.22 (95% CI:1.10–1.35) for respondents without diploma to OR = 0.52 (95% CI:0.47–0.57) for High school +5 or more years level). People from the French overseas departments, immigrants and descendants of immigrants, were all more reluctant to the Covid-19 vaccine (first-generation Africa/Asia immigrants OR = 1.16 (95% CI:1.04–1.30)) *versus* OR = 2.19 (95% CI:1.96–2.43) for the majority population. Finally, our analysis showed that those who reported not trusting the government were more likely to be Covid-19 vaccine-reluctant (OR = 3.29 (95% CI: 3.13–3.45)). Specific campaigns should be thought beforehand to reach women and people at the bottom of the social hierarchy to avoid furthering social inequalities in terms of morbidity and mortality.

## Introduction

Long referred to as the land of Pasteur, France has recently acquired the image of a nation inherently hostile to vaccination, especially since the late 1990's. In 2015, only 52% of French people considered the seasonal flu vaccine to be safe, compared with 85% in the United Kingdom and 80% in Spain [1]. Surveys launched between October and December 2020 confirmed this reputation when it comes to Covid-19 [2]: only 44% of French people were willing to be vaccinated against Covid-19 if they had the opportunity, less than in Germany (65%), Italy (70%), or the United Kingdom (81%), and half as much as in China (91%). France is therefore both one of the countries with the lowest level of acceptance of vaccination in general [3] and

**Data Availability Statement:** All relevant data are within the paper and its Supporting Information files.

**Funding:** NB has received funding from the European Research Council (ERC) under the

European Union's Horizon 2020 research and innovation programme (grant agreement No. [856478]). The funders had no role in study design, data collection and analysis, decision to publish, or preparation of the manuscript.

**Competing interests:** NO authors have competing interests

the Covid-19 vaccine in particular. It makes it an ideal case to study whether the hostility to the Covid-19 vaccine has its own reasons or whether it is related to a reluctance to the principle of vaccination itself.

In addition to the unprecedented and global nature of this pandemic, the rapid development of the vaccine was a first characteristic likely to arouse public distrust [4]. It was, indeed, the first time in the world's vaccine history that a product was developed in such a short time period, less than a year after the first cases. This contrasted dramatically with the last major pandemic, HIV-AIDS, for which, despite the stakes, no vaccine is still available more than three decades after the outbreak. The race for vaccines has resulted in several competing prototypes. The first one to be available on the market, as of December 8, 2020, was developed using messenger RNA technology, which had never before been used as a mode of protection against an epidemic. The introduction of this new technology, whose potential short- and long-term side effects have been widely discussed in the media, may have influenced the willingness to be vaccinated. Another particularity of the Covid-19 vaccine campaign was the strong implication of governments in the procurement of products and in the choice of the prototype. In France, hostility toward the Covid-19 vaccine could be explained by distrust in the government's actions [5] and in foreign pharmaceutical laboratories [6], since no French company produced a vaccine against Covid-19.

These specificities of the Covid-19 vaccination may have had a different impact on vaccination intentions between social groups, which is important to study in order to better target vaccination campaigns.

To study vaccine reluctance, it is important to distinguish vaccine refusal from vaccine hesitancy, defined as "a kind of decision-making process that depends on people's level of commitment to healthism/risk culture and on their level of confidence in the health authorities and mainstream medicine" [7]. Different positions toward vaccination can be articulated: the same individual can be hesitant about vaccines in general but hostile to vaccination against Covid-19, or favorable to vaccines in general but hesitant about vaccination against Covid-19. The challenge here was to account for these different combinations, by correlating them with people's social characteristics.

Our objective was to analyze the social determinants of Covid-19 vaccination reluctance, distinguishing between what related to vaccine distrust in general and what related specifically to the Covid-19 vaccine [8]. The analysis was conducted from an intersectional perspective [9] that simultaneously took into account gender, class, age, and ethno-racial social characteristics, as well as respondents' level of trust in the government.

This study was based on a large-scale random survey of 107,808 people conducted between October 26 and December 9, 2020, a pivotal time, as Pfizer announced on November 9, 2020, that it would be able to produce a 90% effective vaccine on a large scale.

## Materials and methods

### The EpiCoV study

The EpiCoV (Epidémiologie et Conditions de Vie) cohort was set-up in April 2020, with the general aim of understanding the main epidemiological, social and behavioural issues related to the Covid-19 epidemic in France. The survey was approved by the CNIL (French independent administrative authority responsible for data protection) on April 25th 2020 (ref: MLD/MFI/AR205138) and by the "Comité de protection des personnes" (French equivalent of the Research Ethics Committee) on April 24th. The survey also obtained an agreement from the "Comité du Label de la statistique publique", proving its adequacy to statistical quality standards.

A stratified random sample of 135,000 people aged 15 and over, was drawn from the tax database of the National Institute of Statistics and Economic Studies (INSEE), which covers

96% of the population living in France but excludes people living in institutional settings, participated in a first wave of the study in May 2020. People belonging to the lowest decile of income were over-represented. A total of 134,391 respondents participated in the first wave of the study (May 2020). A second wave was conducted in November 2020, including questions on attitudes toward vaccination. Respondents who took part in the first wave of the study were invited to take part in this second wave. In all, 107,808 respondents participated in this second wave (81.7% of the respondents of the first wave of the study). Individuals were invited to answer the questionnaire online, or by phone for those who did not have Internet access. Furthermore, a random sample of 10% of people with Internet access was interviewed by phone in order to take into account a method collection effect. The results published in the study have been adjusted by applying the weights established by the National Institute of Statistics and Economic Studies (INSEE) and marginal recalibration in the survey and sampling design to correct for non-participation, so as to produce estimators that are representative of the population. More information about the cohort can be found in another publication [10].

## Sample information

We focused on people living in metropolitan France, aged 18 and over and likely to decide for themselves whether to be vaccinated (N = 101,112). We chose not to include people who tested positive for Covid-19 (N = 4,036) and whose intention to be vaccinated could be influenced by this fact. Therefore, 85,855 individuals were included in the analysis (Fig 1).

## Outcome measures

To study attitudes toward vaccination in the EpiCoV survey in November 2020, two questions were available. One was about vaccination in general (*Are you strongly*, *somewhat*, *somewhat not*, *or not at all in favor of vaccinations in general*?) and the other was specifically about the Covid-19 vaccine (*If a free vaccine against coronavirus was offered by the Sécurité Sociale* (the French social security system), *would you be willing to get vaccinated*? *Yes probably*, *yes maybe*, *probably not*, *certainly not*, *or you do not know*).

## Social variables

To describe the sample, six sociodemographic variables were considered: age, gender, ethnoracial status (based on migration history), social class (based on current or last occupation), standard of living (based on decile of household income per consumption unit), and formal education level.

Ethno-racial status was defined by combining the criteria of place of birth, nationality, and status of the individual and both parents:

*Majority population*: Persons born in Metropolitan France who are neither first nor second-generation immigrants

*FOD*: Persons or at least one parent born in French Overseas Departments

*First-generation EU*: First-generation immigrants coming from EU27

Second-generation EU: Second-generation immigrants with immigrant parent coming from the EU27

*First generation Africa/Asia*: First-generation immigrants coming from Africa or Asia

*Second-generation Africa/Asia*: Second-generation immigrants with immigrant parent coming from Africa or Asia

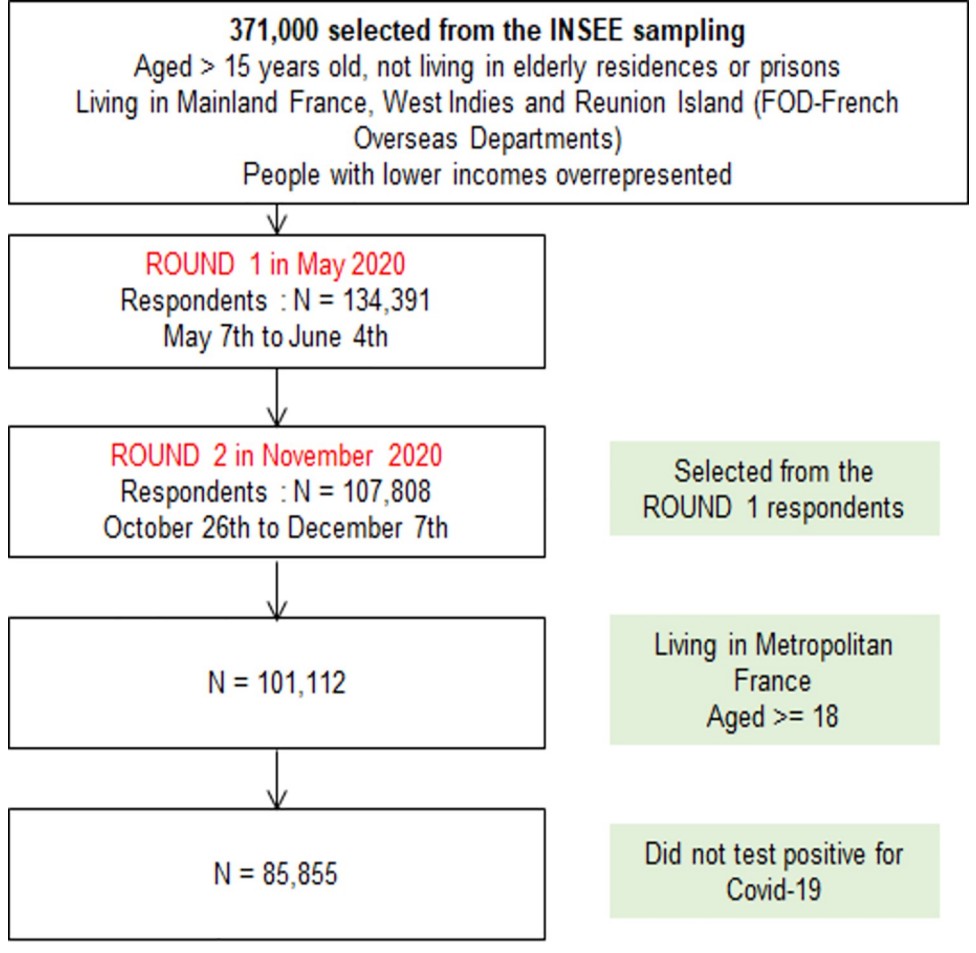

**Fig 1. Flowchart of the national EpiCoV cohort, round 1 (May 2020) and round 2 (November 2020).**

## Statistical analyses

We first described the cross-tabulation of attitudes toward vaccination in general by attitudes toward the Covid-19 vaccine. We then presented the distribution of social characteristics for each attitude toward vaccination in general and toward the Covid-19 vaccine.

We then conducted logistic regressions on being "not at all in favor" to vaccination in general, and on being "certainly not" willing to get vaccinated against Covid-19.

The percentages presented are weighted to account for the sample design. The figures in the tables are not weighted.

All analyses were performed with the R software (1.3.959). A P-value <0.05 was considered statistically significant. Given the sample size, the observed differences were consistently statistically significant. Therefore, no tests are presented for univariable analyses.

## Results

When crossing the question regarding vaccination in general with the question regarding the Covid-19 vaccine, a strong link between the two attitudes emerged, though not without variations (Table 1).

**Table 1. Attitudes toward vaccination in general by attitudes toward the Covid-19 vaccine.**

| | | If a free vaccine against coronavirus was offered by the Sécurité Sociale (the French social security system), would you be willing to get vaccinated? | | | | | |
|---|---|---|---|---|---|---|---|
| | | Yes probably | Yes maybe | Probably not | Certainly not | You do not know | Total |
| *Are you strongly, somewhat, somewhat not, or not at all in favor of vaccinations in general?* | **Strongly in favor** | 16062 (73.2) | 3411 (16.6) | 468 (2.2) | 386 (2.3) | 993 (5.7) | 21320 (100) |
| | **Somewhat in favor** | 12607 (29.9) | 16190 (38.6) | 3901 (8.5) | 2705 (6.7) | 6324 (16.3) | 41727 (100) |
| | **Somewhat not in favor** | 928 (6.4) | 3321 (22.8) | 3947 (25) | 3242 (21.4) | 3408 (24.4) | 14846 (100) |
| | **Not at all in favor** | 227 (3.9) | 524 (7.6) | 1144 (12.7) | 4723 (57.6) | 1344 (18.2) | 7962 (100) |
| **Total** | | 29824 (33.2) | 23446 (27.2) | 9460 (10.3) | 11056 (13.9) | 12069 (15.4) | 85855 (100) |

Almost three quarters of people who were strongly in favor of vaccination in general reported they were willing to be vaccinated against Covid-19. Those who were somewhat in favor of vaccination in general were also more likely to be willing to be vaccinated against Covid-19: more than two thirds of them reported they would perhaps ("Yes probably" or "Yes maybe") get the shot. This was not the case for those who reported they were somewhat not in favor of vaccination in general: 21.4% of them declared they would most likely not get vaccinated against Covid-19, and a quarter of them said they did not know. Those who were not at all in favor of vaccination in general, however, had a stronger position toward the Covid-19 vaccine: more than half of them were determined not to be vaccinated against Covid-19.

Attitudes toward the Covid-19 vaccine seemed to be more definite and socially differentiated (Table 2) than toward vaccination in general (Table 3). Hesitants (those who are "Somewhat in favor" or "Somewhat not in favor" of vaccination in general) made up for more 65% of attitudes toward vaccination in general, whereas only 52% of respondents were unsure of their attitudes toward the Covid-19 vaccine. In both cases, men were more inclined to vaccination than women, and the gender gap was much wider for the Covid-19 vaccine (39.7% of men *versus* 27.3% of women) than for vaccination in general (26.7% of men *versus* and 21.3% of women)

Overall, age played an important role, especially for older adults, but in different ways: in the 25–64 age group, the proportion of people strongly in favor of vaccination in general was around 20% while the age group 65 and over stood out (32.6% were strongly in favor). In the case of the Covid-19 vaccine, the age gradient was very regular from the age of 25 onwards, ranging from 23.3% for the 25–34 age group to 45.1% for people 65 and over.

On the whole, other social characteristics—such as education, social class, and standard of living—played a similar role: the lower in the social hierarchy, the more reluctant one was to vaccination in general and against the Covid-19 vaccine in particular. In both cases, manual workers stood out: 17.1% were not at all in favor of vaccination in general (*versus* 5.9% of the Senior executive professionals) and 17.2% said they would most likely not get vaccinated against Covid19 (*versus* 8.2% of the senior executive professionals).

With regard to ethno-racial status, minorities were always more reluctant to the principle of vaccination, but in different ways: toward vaccination in general, first-generation Africa/Asia immigrants were the most reluctant (27% claimed they were not at all in favor of vaccination in general, compared to 10% in the mainstream population); meanwhile, with regard to the Covid-19 vaccine, it was FOD (French Overseas Departments) natives and descendants of FOD who were the most reluctant (23.7%, compared to 13.3%).

**Table 2. Social characteristics associated with attitudes regarding the Covid-19 vaccine.**

| | Yes probably | Yes maybe | Probably not | Certainly not | You do not know | Total |
|---|---|---|---|---|---|---|
| Total | 29824 (33.2) | 23446 (27.2) | 9460 (10.3) | 11056 (13.9) | 12069 (15.4) | 85855 (100) |
| —————————Sex: | | | | | | |
| Men | 16702 (39.7) | 10865 (27.9) | 3468 (8.7) | 3707 (11) | 4282 (12.7) | 39024 (48) |
| Women | 13122 (27.3) | 12581 (26.5) | 5992 (11.8) | 7349 (16.6) | 7787 (17.8) | 46831 (52) |
| —————————Age: | | | | | | |
| 18–24 | 2892 (31.8) | 2276 (25.8) | 991 (11) | 1440 (18) | 1091 (13.4) | 8690 (10.6) |
| 25–34 | 2428 (23.3) | 2659 (25.4) | 1623 (14.9) | 2237 (22.4) | 1361 (13.9) | 10308 (13.3) |
| 35–44 | 3972 (25.3) | 3971 (26.1) | 2135 (13.8) | 2550 (18.5) | 2151 (16.3) | 14779 (15.7) |
| 45–54 | 5168 (29.3) | 4650 (27.6) | 2029 (11.6) | 2141 (14.4) | 2670 (17.2) | 16658 (16.5) |
| 55–64 | 5747 (33.4) | 4694 (30.2) | 1538 (9.4) | 1534 (10.8) | 2376 (16.3) | 15889 (15.8) |
| + 65 | 9617 (45.1) | 5196 (27.1) | 1144 (5.8) | 1154 (7.3) | 2420 (14.7) | 19531 (28.1) |
| —————————Formal education: | | | | | | |
| No diploma | 1273 (28.8) | 1198 (27.5) | 355 (7.4) | 724 (16.1) | 869 (20.2) | 4419 (10.8) |
| Primary education | 2507 (35.8) | 1911 (27.5) | 587 (7.6) | 746 (10.9) | 1118 (18.1) | 6869 (12.4) |
| Vocational secondary | 5052 (29.7) | 4636 (27) | 1619 (9.2) | 2495 (15.9) | 3110 (18.2) | 16912 (21.1) |
| High school | 5867 (30.7) | 4952 (26.4) | 2273 (12) | 2865 (16.1) | 2648 (14.8) | 18605 (20.8) |
| High school + 2 to 4 years | 8502 (33.1) | 6963 (27.8) | 3158 (12.5) | 3148 (13.5) | 3166 (13.1) | 24937 (23) |
| High school + 5 or more years | 6623 (45.6) | 3786 (26.9) | 1468 (10.6) | 1078 (8.4) | 1158 (8.5) | 14113 (11.9) |
| —————————Social class: | | | | | | |
| Farmers | 430 (35.1) | 348 (27.8) | 121 (8.4) | 140 (10.9) | 188 (17.8) | 1227 (2) |
| Self-employed and entrepreneurs | 1797 (36.7) | 1229 (25.7) | 529 (9.7) | 652 (13.5) | 655 (14.4) | 4862 (6.5) |
| Senior executive professionals | 10216 (46) | 5861 (26.5) | 2103 (9.6) | 1626 (8.2) | 1955 (9.8) | 21761 (18.9) |
| Middle executive professionals | 6065 (33.2) | 5115 (28.7) | 2075 (11.2) | 2230 (13.1) | 2408 (13.8) | 17893 (18.3) |
| Employees | 5871 (26.7) | 6104 (27.2) | 2798 (11.5) | 3649 (16.6) | 4007 (18) | 22429 (27.6) |
| Manual workers | 2632 (27.7) | 2631 (26.9) | 1015 (9.3) | 1667 (17.2) | 1760 (18.9) | 9705 (16.3) |
| Never worked and others | 2813 (33.5) | 2158 (26.6) | 819 (9.4) | 1092 (14.5) | 1096 (15.9) | 7978 (10.5) |
| —————————Standard of living (in deciles): | | | | | | |
| D1 | 1747 (27.5) | 1501 (24.8) | 635 (9.2) | 1117 (19.4) | 1056 (19) | 6056 (8.2) |
| D2-D3 | 2817 (27.1) | 2714 (25.8) | 1234 (10.6) | 1940 (17.9) | 1870 (18.7) | 10575 (18.1) |
| D4-D5 | 3765 (29) | 3659 (26.7) | 1655 (10.7) | 2273 (16.7) | 2204 (16.9) | 13556 (20) |
| D6-D7 | 5705 (31.6) | 5183 (28.6) | 2177 (11.1) | 2499 (13.4) | 2704 (15.4) | 18268 (21.4) |
| D8-D9 | 9198 (38) | 6893 (28.5) | 2633 (10.5) | 2374 (10) | 3059 (12.9) | 24157 (21.9) |
| D10 | 6592 (49.7) | 3496 (26.3) | 1126 (8.3) | 853 (6.7) | 1176 (9) | 13243 (10.5) |
| —————————Ethno-racial status: | | | | | | |
| Majority population | 25375 (34.4) | 19663 (27.3) | 8094 (10.8) | 8823 (13.3) | 9508 (14.3) | 71463 (79.1) |
| Born or parents born in FOD[a] | 243 (23.4) | 242 (25.6) | 129 (12.2) | 225 (23.7) | 157 (15.2) | 996 (1.3) |
| Second generation Europe | 1591 (32.4) | 1251 (27.1) | 480 (9.3) | 634 (14.3) | 749 (16.9) | 4705 (5.6) |
| Second generation Africa/Asia | 751 (24.9) | 667 (22.8) | 327 (10.8) | 588 (21.6) | 541 (19.9) | 2874 (4.1) |
| First generation Europe | 1041 (35.8) | 750 (27.9) | 189 (5.9) | 331 (14) | 419 (16.3) | 2730 (4.1) |
| First generation Africa/Asia | 823 (24.9) | 873 (28) | 241 (7.4) | 455 (14.8) | 695 (25) | 3087 (5.9) |
| ————————— Lives with their children or partner's children: | | | | | | |
| At least a child | 9684 (26.7) | 9220 (26.8) | 4372 (12.3) | 5292 (17.4) | 5111 (16.9) | 33679 (35.5) |
| No child | 20140 (36.9) | 14226 (27.4) | 5088 (9.2) | 5764 (12) | 6958 (14.6) | 52176 (64.5) |
| —————————Regarding the possibility of contracting the virus in the coming months, would you say that you are afraid of contracting it and being seriously ill? | | | | | | |
| Yes | 8842 (42.4) | 5616 (27.6) | 1477 (6.8) | 1607 (8.9) | 2626 (14.2) | 20168 (24.1) |
| No | 20982 (30.3) | 17830 (27) | 7983 (11.4) | 9449 (15.5) | 9443 (15.7) | 65687 (75.9) |

*(Continued)*

**Table 2.** (Continued)

| | Yes probably | Yes maybe | Probably not | Certainly not | You do not know | Total |
|---|---|---|---|---|---|---|
| —————————To limit the spread of the coronavirus, do you trust the government's action?: | | | | | | |
| Yes | 19254 (42.6) | 12656 (29.7) | 3663 (7.8) | 2764 (6.9) | 5064 (12.9) | 43401 (49.2) |
| No | 8777 (24.9) | 8674 (24.7) | 4910 (13.8) | 7117 (22.4) | 4647 (14.3) | 34125 (39.4) |
| You do not know | 1793 (21.5) | 2116 (25) | 887 (9.3) | 1175 (14.8) | 2358 (29.5) | 8329 (11.5) |
| —————————Do you have any COVID comorbidities?: | | | | | | |
| Yes | 10456 (38.1) | 6945 (26.9) | 2288 (8.1) | 2773 (11.7) | 3542 (15.2) | 26004 (33.1) |
| No | 19368 (30.8) | 16501 (27.3) | 7172 (11.4) | 8283 (15) | 8527 (15.4) | 59851 (66.9) |

[a] FOD: French Overseas Departments

Living with a child increased distrust of the vaccine, especially for Covid-19: 17.4% of people living with at least a child responded certainly not to the question on the Covid-19 vaccine (versus 12% of people with no child).

Finally, it was noted that trust in the government was particularly strongly linked to the attitude toward the Covid-19 vaccine, whereas it was somewhat less significant in the case of vaccination in general.

To better understand the specificity of reluctance to vaccinate against Covid 19, we focused on people who expressed their intention to not be vaccinated.

Women appeared to be more reluctant to vaccination in general than men (OR = 1.33 (95% CI: 1.26–1.40)), and even more so with regard to the Covid-19 vaccine (OR = 1.88 (95% CI: 1.79–1.97)), although they were more afraid than men of being infected and being seriously ill, than men, and they took a Covid-19 test more often (S1 Table). The presence of a child was also not equally important according to the type of vaccine: it increased the probability of being hostile to Covid-19 vaccine (OR = 0.12 (95% CI: 1.06–1.18) but was not significant for vaccination in general (Table 4).

The effects of age were also highly significant: the older the respondents were, the less likely they were to be fundamentally hostile to vaccination, although variations remained. It should be noted that while people aged 34 and under were more likely to be reluctant toward Covid-19 vaccine ((OR = 1.32 (95% CI:1.23–1.41)) than toward vaccination in general ((OR = 1.11 (95% CI:1.02–1.21)), the opposite trend was found among those aged 45 and older. Looking at social position, senior executive professionals' attitude is worth highlighting: their attitude toward vaccination in general was not significant, while the probability that they refused the Covid-19 vaccine was lower than that of middle-executive professionals (OR = 0.89 (95% CI: 0.82–0.95)).

A social gradient was found regarding level of education: the higher the degree, the lower the likelihood of being hostile to vaccination, with stronger results for vaccination in general (from OR = 1.53 (95% CI1.38–1.69) for respondents without diploma to OR = 0.39 (95% CI:0.35–0.44) for High school +5 or more years level) than for the Covid-19 vaccine (from OR = 1.22 (95% CI:1.10–1.35) to OR = 0.52 (95% CI:0.47–0.57) for the same levels).

Vaccine reluctance was also related to financial resources. Being in the lowest deciles increased the odds of being fundamentally hostile to vaccination in general (from OR = 1.16 (95% CI: 1.08–1.28) down to OR = 0.69 (95% CI: 0.62–0.77)) for the richest; same trend was observed for the Covid-19 vaccine.

The ethno-racial status played an important role. People who did not belong to the so-called mainstream population, *i.e.*, those from the French DOM, immigrants and descendants of immigrants from Africa/Asia, were all more reluctant to vaccination. Interestingly, hostility to

**Table 3. Social characteristics associated with attitudes regarding vaccination in general.**

| | Strongly in favor | Somewhat in favor | Somewhat not in favor | Not at all in favor | Total |
|---|---|---|---|---|---|
| Total | 21320 (23.9) | 41727 (47.6) | 14846 (17.3) | 7962 (11.2) | 85855 (100) |
| ——————Sex: | | | | | |
| Men | 11175 (26.7) | 18599 (47.2) | 6010 (15.9) | 3240 (10.2) | 39024 (48) |
| Women | 10145 (21.3) | 23128 (48) | 8836 (18.6) | 4722 (12.1) | 46831 (52) |
| ——————Age: | | | | | |
| 18–24 | 2253 (24.4) | 4293 (48.9) | 1423 (16.8) | 721 (10) | 8690 (10.6) |
| 25–34 | 2145 (19.6) | 4894 (46.6) | 2065 (19.9) | 1204 (13.9) | 10308 (13.3) |
| 35–44 | 3026 (19) | 7549 (49.3) | 2668 (18.7) | 1536 (13) | 14779 (15.7) |
| 45–54 | 3462 (19.5) | 8343 (48.2) | 3164 (19.7) | 1689 (12.6) | 16658 (16.5) |
| 55–64 | 3589 (20.8) | 8051 (50.1) | 2845 (18.5) | 1404 (10.5) | 15889 (15.8) |
| + 65 | 6845 (32.6) | 8597 (44.9) | 2681 (13.5) | 1408 (9) | 19531 (28.1) |
| ——————Formal education: | | | | | |
| No diploma | 831 (20) | 2025 (44.9) | 794 (16.4) | 769 (18.7) | 4419 (10.8) |
| Primary education | 1663 (24.7) | 3373 (48.3) | 1118 (15.6) | 715 (11.4) | 6869 (12.4) |
| Vocational secondary | 3053 (18.4) | 8225 (47.5) | 3314 (19.2) | 2320 (14.9) | 16912 (21.1) |
| High school | 4002 (21.6) | 9141 (47.9) | 3585 (19.2) | 1877 (11.2) | 18605 (20.8) |
| High school + 2 to 4 years | 6233 (24.7) | 12585 (49.6) | 4374 (17.9) | 1745 (7.8) | 24937 (23) |
| High school + 5 or more years | 5538 (38.3) | 6378 (45.2) | 1661 (12.2) | 536 (4.3) | 14113 (11.9) |
| ——————Social class: | | | | | |
| Farmers | 263 (22.1) | 616 (50.7) | 222 (16.3) | 126 (10.9) | 1227 (2) |
| Self-employed and entrepreneurs | 1085 (21.9) | 2265 (45.9) | 950 (19.4) | 562 (12.8) | 4862 (6.5) |
| Senior executive professionals | 7704 (35.3) | 10088 (45.3) | 2864 (13.5) | 1105 (5.9) | 21761 (18.9) |
| Middle executive professionals | 4346 (24.2) | 8889 (48.8) | 3238 (18) | 1420 (9) | 17893 (18.3) |
| Employees | 4052 (18.8) | 11264 (48.7) | 4517 (19.5) | 2596 (13) | 22429 (27.6) |
| Manual workers | 1551 (17.4) | 4713 (47.2) | 1910 (18.3) | 1531 (17.1) | 9705 (16.3) |
| Never worked and others | 2319 (27.5) | 3892 (47.9) | 1145 (14.6) | 622 (10) | 7978 (10.5) |
| ——————Standard of living (in deciles): | | | | | |
| D1 | 1269 (20.1) | 2748 (44.1) | 1152 (18.3) | 887 (17.5) | 6056 (8.2) |
| D2-D3 | 1979 (19.4) | 5012 (46.1) | 2104 (18.9) | 1480 (15.5) | 10575 (18.1) |
| D4-D5 | 2528 (19.8) | 6743 (48.2) | 2667 (18.9) | 1618 (13.2) | 13556 (20) |
| D6-D7 | 3959 (22.4) | 9222 (49.5) | 3395 (17.9) | 1692 (10.1) | 18268 (21.4) |
| D8-D9 | 6503 (27.1) | 12071 (49.1) | 3892 (16) | 1691 (7.7) | 24157 (21.9) |
| D10 | 5082 (38.4) | 5931 (44.7) | 1636 (12.2) | 594 (4.7) | 13243 (10.5) |
| ——————Ethno-racial status: | | | | | |
| Majority population | 17924 (24.2) | 35201 (48.5) | 12221 (17.2) | 6117 (10.1) | 71463 (79.1) |
| Born or parents born in FOD[a] | 194 (18.6) | 457 (45) | 191 (18.4) | 154 (18) | 996 (1.3) |
| Second generation Europe | 1132 (22.9) | 2261 (48) | 870 (18.7) | 442 (10.5) | 4705 (5.6) |
| Second generation Africa/Asia | 585 (19.8) | 1315 (44.6) | 560 (19.1) | 414 (16.4) | 2874 (4.1) |
| First generation Europe | 840 (29.1) | 1200 (43.7) | 412 (14.9) | 278 (12.4) | 2730 (4.1) |
| First generation Africa/Asia | 645 (20.4) | 1293 (41) | 592 (17.9) | 557 (20.7) | 3087 (5.9) |
| ——————Lives with their children or partner's children: | | | | | |
| At least a child | 6987 (19.3) | 17033 (49) | 6200 (18.8) | 3459 (12.8) | 33679 (35.5) |
| No child | 14333 (26.4) | 24694 (46.8) | 8646 (16.5) | 4503 (10.3) | 52176 (64.5) |

[a] FOD: French Overseas Departments

**Table 4. Factors associated with vaccination in general and Covid-19 vaccine refusals.**

| | Covid vaccine: Certainly not | | | | Vaccination in general: Not at all in favor | | | |
|---|---|---|---|---|---|---|---|---|
| | Frequency | OR | 95% IC | p-value | Frequency | OR | 95% IC | p-value |
| Total | 13.9 (85855) | | | | 11.2 (85855) | | | |
| —————————Sex: | | | | | | | | |
| Men (ref.) | 11 (39024) | 1 | | <0.0001 | 10.2 (39024) | 1 | | <0.0001 |
| Women | 16.6 (46831) | **1.88** | **[1.79–1.97]** | | 12.1 (46831) | **1.33** | **[1.26–1.40]** | |
| —————————Age: | | | | | | | | |
| 18–24 | 18 (8690) | 1.05 | [0.95–1.16] | <0.0001 | 10 (8690) | **0.74** | **[0.65–0.84]** | <0.0001 |
| 25–34 | 22.4 (10308) | **1.32** | **[1.23–1.41]** | | 13.9 (10308) | **1.11** | **[1.02–1.21]** | |
| 35–44 (ref.) | 18.5 (14779) | 1 | | | 13 (14779) | 1 | | |
| 45–54 | 14.4 (16658) | **0.73** | **[0.68–0.78]** | | 12.6 (16658) | 0.95 | [0.88–1.03] | |
| 55–64 | 10.8 (15889) | **0.59** | **[0.54–0.63]** | | 10.5 (15889) | **0.83** | **[0.76–0.90]** | |
| + 65 | 7.3 (19531) | **0.4** | **[0.36–0.43]** | | 9 (19531) | **0.72** | **[0.66–0.79]** | |
| —————————Formal education: | | | | | | | | |
| No diploma | 16.1 (4419) | **1.22** | **[1.10–1.35]** | <0.0001 | 18.7 (4419) | **1.53** | **[1.38–1.69]** | <0.0001 |
| Primary education | 10.9 (6869) | 1 | [0.91–1.10] | | 11.4 (6869) | **1.23** | **[1.11–1.35]** | |
| Vocational secondary | 15.9 (16912) | **1.15** | **[1.08–1.23]** | | 14.9 (16912) | **1.41** | **[1.31–1.51]** | |
| High school (ref.) | 16.1 (18605) | 1 | | | 11.2 (18605) | 1 | | |
| High school + 2 to 4 years | 13.5 (24937) | **0.81** | **[0.77–0.87]** | | 7.8 (24937) | **0.69** | **[0.64–0.75]** | |
| High school + 5 or more years | 8.4 (14113) | **0.52** | **[0.47–0.57]** | | 4.3 (14113) | **0.39** | **[0.35–0.44]** | |
| —————————Social class: | | | | | | | | |
| Farmers | 10.9 (1227) | 1.1 | [0.91–1.34] | <0.0001 | 10.9 (1227) | 1.13 | [0.92–1.38] | <0.0001 |
| Self-employed and entrepreneurs | 13.5 (4862) | **1.18** | **[1.07–1.30]** | | 12.8 (4862) | **1.28** | **[1.15–1.43]** | |
| Senior executive professionals | 8.2 (21761) | **0.88** | **[0.82–0.95]** | | 5.9 (21761) | 1.01 | [0.92–1.10] | |
| Middle executive professionals (ref.) | 13.1 (17893) | 1 | | | 9 (17893) | 1 | | |
| Employees | 16.6 (22429) | 1.02 | [0.96–1.09] | | 13 (22429) | **1.09** | **[1.01–1.17]** | |
| Manual workers | 17.2 (9705) | **1.14** | **[1.05–1.23]** | | 17.1 (9705) | **1.28** | **[1.18–1.40]** | |
| Never worked and others | 14.5 (7978) | **0.66** | **[0.60–0.73]** | | 10 (7978) | **0.7** | **[0.62–0.80]** | |
| —————————Standard of living (in deciles): | | | | | | | | |
| D1 | 19.4 (6056) | 1.07 | [0.98–1.16] | <0.0001 | 17.5 (6056) | **1.16** | **[1.06–1.28]** | <0.0001 |
| D2-D3 | 17.9 (10575) | 1.02 | [0.95–1.09] | | 15.5 (10575) | 1.07 | [0.99–1.16] | |
| D4-D5 (ref.) | 16.7 (13556) | 1 | | | 13.2 (13556) | 1 | | |
| D6-D7 | 13.4 (18268) | 0.93 | [0.88–1] | | 10.1 (18268) | **0.9** | **[0.83–0.97]** | |
| D8-D9 | 10 (24157) | **0.81** | **[0.76–0.87]** | | 7.7 (24157) | **0.82** | **[0.76–0.88]** | |
| D10 | 6.7 (13243) | **0.69** | **[0.63–0.76]** | | 4.7 (13243) | **0.69** | **[0.62–0.77]** | |
| —————————Ethno-racial status: | | | | | | | | |
| Majority population | 13.3 (71463) | 1 | | <0.0001 | 10.1 (71463) | 1 | | <0.0001 |
| Born or parents born in FOD[a] | 23.7 (996) | **1.66** | **[1.41–1.95]** | | 18 (996) | **1.74** | **[1.45–2.08]** | |
| Second generation Europe | 14.3 (4705) | **1.17** | **[1.06–1.28]** | | 10.5 (4705) | 1.07 | [0.96–1.19] | |
| Second generation Africa/Asia | 21.6 (2874) | **1.36** | **[1.23–1.51]** | | 16.4 (2874) | **1.61** | **[1.44–1.80]** | |
| First generation Europe | 14 (2730) | **1.16** | **[1.03–1.31]** | | 12.4 (2730) | **1.28** | **[1.12–1.46]** | |
| First generation Africa/Asia | 14.8 (3087) | **1.16** | **[1.04–1.30]** | | 20.7 (3087) | **2.19** | **[1.96–2.43]** | |
| —————————Lives with their children or partner's children: | | | | | | | | |
| At least a child | 17.4 (33679) | **1.12** | **[1.06–1.18]** | <0.0001 | 12.8 (33679) | 0.95 | [0.89–1] | 0.07242 |
| No child (ref.) | 12 (52176) | 1 | | | 10.3 (52176) | 1 | | |
| —————————Regarding the possibility of contracting the virus in the coming months, would you say that you are afraid of contracting it and being seriously ill? | | | | | | | | |
| Yes | 8.9 (20168) | **0.57** | **[0.54–0.61]** | <0.0001 | 7.4 (20168) | **0.57** | **[0.53–0.60]** | <0.0001 |

*(Continued)*

**Table 4.** (Continued)

| | Covid vaccine: Certainly not | | | | Vaccination in general: Not at all in favor | | | |
|---|---|---|---|---|---|---|---|---|
| | Frequency | OR | 95% IC | p-value | Frequency | OR | 95% IC | p-value |
| No (ref.) | 15.5 (65687) | 1 | | | 12.4 (65687) | 1 | | |
| ————————To limit the spread of the coronavirus, do you trust the government's action?: | | | | | | | | |
| Yes (ref.) | 6.9 (43401) | 1 | | <0.0001 | 6.5 (43401) | 1 | | <0.0001 |
| No | 22.4 (34125) | **3.29** | **[3.13–3.45]** | | 15.5 (34125) | **2.68** | **[2.54–2.83]** | |
| You do not know | 14.8 (8329) | **1.87** | **[1.74–2.02]** | | 17.1 (8329) | **2.31** | **[2.14–2.49]** | |
| ————————Do you have any COVID comorbidities?: | | | | | | | | |
| Yes | 11.7 (26004) | **0.89** | **[0.84–0.93]** | <0.0001 | 10.5 (26004) | **0.88** | **[0.84–0.93]** | <0.0001 |
| No (ref.) | 15 (59851) | 1 | | | 11.6 (59851) | 1 | | |

n = 85855, OR: Odds Ratio,Parameters with a significant odds ratio compared to the reference are in bold.

The regressions were also adjusted on the week of completion of the questionnaire (not shown).

[a] FOD: French Overseas Departments

the Covid-19 vaccine remained higher than that of the mainstream population, but the differences were less marked (for first-generation Africa/Asia immigrants OR = 1.16 (95% CI:1.04–1.30)) *versus* OR = 2.19 (95% CI:1.96–2.43)).

Attitudes toward vaccination also depended on a person's perception of both the disease and the officials in charge of the vaccination policies. As expected, fear of the disease made people less likely to belong to the Covid-19 vaccine-reluctant group (OR = 0.57 (95% CI: 0.54–0.61)). The link between trust in the government and trust in vaccination should also be highlighted: those who reported not trusting the government were more likely to be Covid-19 vaccine-reluctant (OR = 3.29 (95% CI: 3.13–3.45)) and more likely to be "not at all in favor" of vaccination in general (OR = 2.68(95% CI: 2.54–2.83)) than those who reported trusting the government.

## Discussion

The EpiCoV survey is the first national randomized socio-epidemiological survey of this scale to study the specificity of the response to Covid-19 vaccination, taking into account gender, class, age, and ethno-racial characteristics [11] as well as level of trust in government actions.

Our results showed that Covid-19 vaccine hesitancy was highly, but not totally, correlated with hostility to vaccination in general and had specific social determinants. Based on the distinction between vaccine refusal and hesitancy [12], our analyses highlighted the need to consider reluctants as a specific group, distinct from the hesitants, contrary to what has been done in some recent work [13, 14]. Respondents at the bottom of the social hierarchy were more likely to be reluctant toward Covid-19 vaccination, but to a lesser extent than toward vaccination in general. An important gender specificity was found: women were much more reluctant toward Covid-19 vaccination than toward vaccination in general. Our analyses also showed that trust in government was the variable with the strongest association with reluctance to vaccination against Covid-19, even stronger than for the vaccine in general.

Although France is a country where the prevalence of vaccine reluctance is particularly high, the social characteristics of French people hostile to the Covid-19 vaccine are comparable to those found in other countries. First of all, our results confirmed women's greater reluctance to COVID-19 vaccination, already observed in other surveys in France [14, 15], in the United Kingdom, in China and in the United States [16, 17]. Though many studies have been able to measure women's higher reluctance toward the Covid-19 vaccine, only few explanations were

provided. First, it should be noted that women's critical discourse toward vaccination is long-standing and already widely documented: in the 1970s and 1980s, women's movements in the United States demanded more accurate information and transparency from the government regarding injections offered to their children [18]. Few years ago, a study on anti-vaccination mobilizations on Facebook networks in Australia and North America revealed the very strong presence of women in these activist groups [19]. Women's greater reluctance to vaccination could also be linked to their "cultural health capital" [20], which reflects a gendered bound to the body, partly resulting from a different socialization process of women and men regarding pain and health [21]. The inclination toward complementary and alternative medicine [22] could thus explain women's greater reluctance to resort to medical practices over which they have no control.

Our results showed for the first time that this gendered reluctance toward Covid-19 vaccine was much stronger than toward vaccination in general.

At first glance, one could assume that it reflects a reasoned anticipation of the risk of complications. Men were proven to be more likely to contract severe forms of the disease; therefore, women could rightly consider themselves less exposed to the lethality of Covid-19 and therefore less concerned by the need for the vaccine. However, this was not the case. Women were more apprehensive about the disease: according to our survey, they were more afraid than men of contracting the virus and being seriously ill and they took a Covid-19 test more often (S1 Table). If we ruled out the idea that women were less afraid of contracting serious forms, three specific Covid-19 reluctance hypotheses could be formulated.

The first hypothesis was that the vaccine against Covid-19 could pose a threat to maternity plans. In the 25–34 age group, women were more hostile to vaccines in general and even more so to the Covid-19 vaccine. At an age range that is socially devoted to motherhood, women were more concerned about the possible effects an injection of a very recently-developed product in their body could have on a possible pregnancy. This reluctance could be linked to their greater aversion to childhood vaccination [23], as they consider that the intensive mothering practices (feeding, nutrition and natural living) they provided to their children would be preferable to external medical protection [24], thus preferring natural immune defenses over those offered by vaccination. In contrast, as of age 45, the probability of women refusing to be vaccinated against Covid-19 decreased continuously as age increased, which was not the case with vaccination in general. Once past the social age of motherhood, the fear associated with the consequences of a Covid-19 vaccine injection faded, supporting the hypothesis of gendered reluctance at maternal ages.

The second explanation could be found in the relationship that women have to their social role as caregivers within the family [25]. It was probably for this reason that women living with a child were, regardless of age, even more reluctant to the new vaccine than to vaccines in general [26]. Moreover, getting the Covid-19 vaccine could appear both as a medical intervention and as an external interference in the domestic sphere. Thus, the assignment of women to domestic tasks may have made them more reluctant than men to accept governmental interference, particularly marked for the Covid-19 vaccine, in the private sphere.

The third hypothesis involved a gendered relationship to health and environmental risks, which is also the product of primary socializations [27]. In the case of Covid-19, the large-scale distribution of a messenger RNA vaccine, which was at the centre of daily media debates in November 2020, may have been a greater concern for women than for men because of their stronger aversion to technology-related risks [28]. In the short term, it constituted a guarantee of being protected against Covid-19 for all those who would have benefited from an injection. However, there was still some uncertainty about the long-term effects that could emerge on cell transformation if this type of vaccine were to be generalized every year over a long period

of time and to the entire population. The greater reluctance of women to receive the Covid-19 vaccine might have been due to a differentiated socialization making them more sensitive than men to long-term risks that could have a profound effect on the body and health. Conversely, men's greater inclination toward the Covid-19 vaccine might also have been the result of greater acceptance of technological innovations in genetics [29].

Our survey also showed that reluctance toward Covid-19 vaccination was closely related to other demographic and socioeconomic characteristics.

As they got older, respondents were less likely to refuse vaccination. But this age effect was even more pronounced for the vaccination against Covid-19, reflecting the fact that older people were much more likely to experience serious complications if contracting the virus.

Respondents with lower levels of education were more likely to be reluctant toward vaccination in general and, to a lesser extent, toward the Covid-19 vaccine. This distrust is partly explained by the fact that members of the working classes have a perception of their body and their health which is more distant from medical diagnoses than in upper classes. The lower magnitude of the social gradient for Covid-19 vaccination may be due to the pandemic and uncertain nature of the disease, which affects all social groups.

The marked income gradient for vaccination in general, as well as for vaccination against Covid-19, even though vaccination is free in France, may reflect the fact that the loss of income in the event of illness would be more important for the rich than for the poor.

Ethno-racial minorities appeared to be more hostile than the majority population to vaccination in general which confirmed studies on the greater reluctance of African Americans in the United States to receive the new vaccine [13]. Numerous studies have shown that ethno-racial minorities have less confidence in the healthcare system and in caregivers than the majority population [30–32]. In the case of France, this mistrust can be explained on the one hand by the weight of its colonial history and the associated pharmaceutical scandals [33], and on the other by discrimination and mistreatment to which these populations may have been exposed when resorting to the public health system [32]. Interestingly, their hostility to the Covid-19 vaccine was less marked than for the vaccine in general. As immigrants and descendants of immigrants from Africa are more often affected by the disease [32], it is likely that this lesser hostility reflects a greater effective proximity to the disease.

Finally, our analyses showed that trust in government was the variable with the strongest effect on reluctance to vaccinate against Covid-19, even stronger than for the vaccine in general. These results confirmed a link between vaccine adherence and trust in government, demonstrated prior to the Covid-19 pandemic [5]. In a country such as France, public authorities have close control over the supply and marketing of vaccines. Therefore, people's propensity to trust the government, leading actor in the country's vaccination strategy [5], affected attitudes toward vaccination. The French government has been harshly criticized for failing to anticipate the crisis and for wanting to cover up the lack of masks, claiming until April 2020 that they were not necessary to protect oneself from the virus. The link between confidence in the government—or being close to the governing parties [5]—and vaccination intention was also strengthened when comparing vaccine supplies available in other countries: the vaccination rate in the United States, Israel and other European nations has fueled a feeling of downgrading, undoubtedly deteriorating the citizens' level of confidence in their government and in its ability to lead a successful vaccination campaign.

It should be added that during the period of the survey, the vaccination campaign was still being developed by French authorities. In the course of November 2020, the main announcement from officials was made from the French President on November 24th, to announce that the vaccination campaign would be "swift" and "massive", but that getting vaccinated would not be compulsory [34]. It would be coherent to assume that the announcement did not

influence the attitude towards the Covid-19 vaccine, and that the link between confidence in the government and reluctance to getting vaccinated was developed before the survey period.

Like all national surveys conducted in the general population, our analysis showed limitations. First, the study could not reach highly vulnerable groups such as the undocumented and the homeless, who were particularly affected by the pandemic [35], especially in France [36]. Furthermore, our analyses did not take into account which sources of information people used to learn about vaccination issues. Misinformation campaigns in the media and on social networks could have influenced vaccination intentions [37, 38]. However, the impact of these discourses were not homogeneous and it could be hypothesized that their effects varied according to social background and gender, somehow reinforcing the results we have obtained. Finally, it is important to note that the survey was conducted shortly before the vaccines were actually made available in France in early January 2021. Attitudes toward vaccination might have changed according to available information on each prototype vaccine and might as well have changed over time [39]. As the number of vaccinated individuals increased, knowing vaccinated people in one's environment might encourage reluctant individuals to follow suit. However, the scarce studies on the evolution of vaccination intentions over time showed that it was mainly those who were hesitant who were likely to be vaccinated [40]. In the case of France, available data showed that the share of clearly reluctant individuals, those on whom we focused our analyses, remained relatively stable over time between July 2020 and February 2021 [41, 42].

Finally, our results suggest that the vaccination strategy used in France should be reconsidered. It is based exclusively on epidemiological criteria, with priority access to vaccines being reserved initially for the oldest or those with comorbidities. Some groups will be more difficult to convince than others in the vaccination campaign: women, youth, working class, ethnoracial minorities. Specific campaigns should be thought beforehand to reach these people. In particular, ethno-racial minorities are both more exposed to this pandemic and more reluctant to be vaccinated than the majority population, so a major effort must be made to reach them in this vaccination campaign. Failure to take into account the social determinants of reluctance to vaccinate could lead to strengthening social inequalities in terms of morbidity and mortality [43, 44], as well as in terms of care work, mental health, and sexual and reproductive health, which particularly affect women [45].

## Supporting information

**S1 Table. Covid-19 test and scare of contracting the virus and being seriously ill, according to sex.**
(DOCX)

## Acknowledgments

The authors warmly thank all the volunteers of the EpiCov cohort*; the DREES and INSEE teams; the staff of IPSOS, Inserm Santé Publique team and Frédéric Robergeau.

*The EPICOV study group: Nathalie Bajos (co-principal investigator), Josiane Warszawski (co-principal investigator), Guillaume Bagein, François Beck, Emilie Counil, Florence Jusot, Nathalie Lydie, Claude Martin, Laurence Meyer, Philippe Raynaud, Alexandra Rouquette, Ariane Pailhé, Delphine Rahib, Patrick Sicard, Rémy Slama, Alexis Spire.

## Author Contributions

**Conceptualization:** Nathalie Bajos, Alexis Spire.

**Formal analysis:** Nathalie Bajos, Alexis Spire, Léna Silberzan.

**Funding acquisition:** Nathalie Bajos.

**Investigation:** Nathalie Bajos.

**Methodology:** Nathalie Bajos.

**Project administration:** Nathalie Bajos.

**Resources:** Nathalie Bajos.

**Supervision:** Nathalie Bajos, Alexis Spire.

**Validation:** Nathalie Bajos, Alexis Spire.

**Writing – original draft:** Nathalie Bajos, Alexis Spire.

**Writing – review & editing:** Nathalie Bajos, Alexis Spire.

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
