## [Decision Letter · Decision Letter 0]

23 Nov 2021

PONE-D-21-30412The social specificities of hostility toward vaccination against Covid-19 in FrancePLOS ONE

Dear Dr. Bajos,

Thank you for submitting your manuscript to PLOS ONE. After careful consideration, we feel that it has merit but does not fully meet PLOS ONE’s publication criteria as it currently stands. Therefore, we invite you to submit a revised version of the manuscript that addresses the points raised during the review process.

We look forward to receiving your revised manuscript.

Kind regards,

Sanjay Kumar Singh Patel, Ph.D.

Academic Editor

PLOS ONE

Journal Requirements:

“NB has received funding from the European Research Council (ERC) under the European Union’s Horizon 2020 research and innovation programme (grant agreement No. [856478])”

5. We note you have included a table to which you do not refer in the text of your manuscript. Please ensure that you refer to Table 3 in your text; if accepted, production will need this reference to link the reader to the Table.

Reviewers' comments:

Reviewer's Responses to Questions

**Comments to the Author**

1. Is the manuscript technically sound, and do the data support the conclusions?

Reviewer #1: Yes

Reviewer #2: Yes

2. Has the statistical analysis been performed appropriately and rigorously? 

Reviewer #1: Yes

Reviewer #2: Yes

3. Have the authors made all data underlying the findings in their manuscript fully available?

Reviewer #1: Yes

Reviewer #2: Yes

4. Is the manuscript presented in an intelligible fashion and written in standard English?

Reviewer #1: Yes

Reviewer #2: Yes

5. Review Comments to the Author

Reviewer #1: The manuscript is well written, simple, structured language used, and properly organized. Other than some minor problem mentioned in the comments, this research article is quite interesting and easy to understand.

1. In Page 5, Since the cohort procedure is published in another publication, the phrase "Since the cohort protocol is detailed in another publication," requires a bit more background information.

2. In page 6, It would be better to include the number of people who tested positive for Covid-19 (N=4,036) and whose intention to be vaccinated could be influenced by this fact in the statement "We chose not to include people who tested positive for Covid-19 (N=4,036) and whose intention to be vaccinated could be influenced by this fact."

3. On Page 10, insert a square bracket"]" in the statement "The third hypothesis involved a gendered relationship to health and environmental risks, which is also the product of primary socializations [29."

4. Modify and correct the phrase structure of the statement "The older people were, the less they refused vaccination" on page 11.

5. In page 13, Instead of giving a direct link, such as "“(https://www.santepubliquefrance.fr/etudes-et-enquetes/coviprev-une-enquete-pour suivre-l-evolution-des-comportements-et-de-la-sante-mentale-pendant-l-epidemie-de covid- 19#block-249162)” it would be better to make it a reference and cite in.

6. Few references in the text are from 2004 and 2010, and efforts should be made to provide relevant recent references that are no more than five years old.

Reviewer #2: In the current research article entitled "The social specificities of hostility toward vaccination against Covid-19 in France", by Bajos et al., have studied/surveyed opposition toward vaccination against Covid-19 in France. They found that, in France a gendered reluctance toward vaccination in general but even more so regarding vaccination against COVID-19. Further, not only people at the bottom of the social hierarchy but also immigrants and descendants of immigrants were all more reluctant to the Covid-19 vaccine. This article addresses a research topic of great interest, which is under intense investigation in the past 2 years and the manuscript is generally well-written. However, this reviewer has certain suggestions that would help produce a more comprehensive overview of the topic:

Suggestions:

1. At least one additional Figure (illustration) may be provided as to highlight the summary or prospect of this study.

2. One paragraph can be added to discussion the Government of France efforts to improve COVID-19 vaccination during the survey period.

---

## [Author Response · Author response to Decision Letter 0]

16 Dec 2021

We would like to sincerely thank the reviewers for their helpful comments, which helped us improve the manuscript. 

Journal Requirements:

The manuscript was modified so as to meet PLOS ONE’s style requirements.

“NB has received funding from the European Research Council (ERC) under the European Union’s Horizon 2020 research and innovation programme (grant agreement No. [856478])”

"The funders had no role in study design, data collection and analysis, decision to publish, or preparation of the manuscript." added to the text in the “Funding details” statement and to the Cover letter. 

Data availability 

Data of the study are protected under the protection of health data regulation set by the French National Commission on Informatics and Liberty (Commission Nationale de l’Informatique et des Libertés, CNIL) in line with the European regulations and the Data Protection Act .

The data can be available upon reasonable request to the co-principal investigator of the study (josiane.warszawski@inserm.fr). The French law forbids us to provide free access to EPICOV data; access could however be given by the EPICOV steering committee after legal verification of the use of the data. Please, feel free to come back to us should you have any additional questions.

A supporting information file was added, as well as a sentence in the Results section.

Women appeared to be more reluctant to vaccination in general than men (OR=1.33 (95% CI: 1.26-1.40)), and even more so with regard to the Covid-19 vaccine (OR=1.88 (95% CI: 1.79-1.97)), although they were more afraid than men of being infected and being seriously ill, than men, and they took a Covid-19 test more often (S1 Fig).

In the Discussion session, reference was made to S1 Fig.

5. We note you have included a table to which you do not refer in the text of your manuscript. Please ensure that you refer to Table 3 in your text; if accepted, production will need this reference to link the reader to the Table.

We included the reference to Table 3 in the following sentence: “The presence of a child was also not equally important according to the type of vaccine: it increased the probability of being hostile to Covid-19 vaccine (OR=0.12 (95% CI: 1.06-1.18) but was not significant for vaccination in general (Table 3)”

Following Reviewer #1’s remarks on references cited in the manuscript, a couple of references were removed due to the publication date, and some were replaced by more current research. 

● We removed the following: 

1. Bertrand A, Torny D. Libertés individuelles et santé collective : Une étude socio-historique de l’obligation vaccinale. Convention CNRS/ DGS SD5C 03-673: CERMES CNRS UMR 8559 – INSERM U502 – EHESS; 2004 p. 109.

32. Boltanski L. Les usages sociaux du corps*. Annales Histoire, Sciences Sociales. 1971;26: 205–233. doi:10.3406/ahess.1971.422470

37. Gamble VN. Under the shadow of Tuskegee: African Americans and health care. Am J Public Health. 1997;87: 1773–1778.

38. Papon S, Robert-Bobée I. Une hausse des décès deux fois plus forte pour les personnes nées à l’étranger que pour celles nées en France en mars-avril 2020. Insee Focus. 2020 [cited 16 Apr 2021]. Available: https://www.insee.fr/fr/statistiques/4627049

39. Salmon DA, Moulton LH, Omer SB, DeHart MP, Stokley S, Halsey NA. Factors associated with refusal of childhood vaccines among parents of school-aged children: a case-control study. Arch Pediatr Adolesc Med. 2005;159: 470–476. doi:10.1001/archpedi.159.5.470

40. Marlow LAV, Waller J, Wardle J. Parental attitudes to pre-pubertal HPV vaccination. Vaccine. 2007;25: 1945–1952. doi:10.1016/j.vaccine.2007.01.059

28. Cayouette-Remblière J, Lambert A. L’explosion des inégalités. Classes, genre et générations face à la crise sanitaire. L’aube. 2021

● We replaced the following:

21. Shim JK. Cultural health capital: A theoretical approach to understanding health care interactions and the dynamics of unequal treatment. J Health Soc Behav. 2010;51: 1–15. doi:10.1177/0022146509361185

(replaced by 

21. Dubbin LA, Chang JS, Shim JK. Cultural health capital and the interactional dynamics of patient-centered care. Soc Sci Med. 2013;93: 10.1016/j.socscimed.2013.06.014. doi:10.1016/j.socscimed.2013.06.014)

27. Benin AL, Wisler-Scher DJ, Colson E, Shapiro ED, Holmboe ES. Qualitative Analysis of Mothers’ Decision-Making About Vaccines for Infants: The Importance of Trust. Pediatrics. 2006;117: 1532–1541. doi:10.1542/peds.2005-1728

(replaced by

27. Dubé E, Vivion M, MacDonald N. Vaccine hesitancy, vaccine refusal and the anti-vaccine movement: influence, impact and implications. Expert Rev Vaccines. 2014;14: 99–117. doi:10.1586/14760584.2015.964212)

30. Greenberg MR, Schneider DF. Gender differences in risk perception: effects differ in stressed vs. non-stressed environments. Risk Anal. 1995;15: 503–511. doi:10.1111/j.1539-6924.1995.tb00343.x

(replaced by 

30. Maxfield S, Shapiro M, Gupta V, Hass S. Gender and risk: women, risk taking and risk aversion. Gender in Management: An International Journal. 2010;25: 586–604. doi:10.1108/17542411011081383)

● In-text citations were changed as such: 

Page 11, reference in the following “[...] and on the other by discrimination and mistreatment to which these populations may have been exposed when resorting to the public health system” was changed from 

37. Gamble VN. Under the shadow of Tuskegee: African Americans and health care. Am J Public Health. 1997;87: 1773–1778.

to

35. Kazemian S, Fuller S, Algara C. The role of race and scientific trust on support for COVID-19 social distancing measures in the United States. PLOS ONE. 2021;16: e0254127. doi:10.1371/journal.pone.0254127

Page 12, reference in the following sentence “These results confirmed a link between vaccine adherence and trust in government, demonstrated prior to the Covid-19 pandemic” was changed from 

38. Papon S, Robert-Bobée I. Une hausse des décès deux fois plus forte pour les personnes nées à l’étranger que pour celles nées en France en mars-avril 2020. Insee Focus. 2020 [cited 16 Apr 2021]. Available: https://www.insee.fr/fr/statistiques/4627049

to 

 Ward JK, Alleaume C, Peretti-Watel P, Peretti-Watel P, Seror V, Cortaredona S, et al. The French public’s attitudes to a future COVID-19 vaccine: The politicization of a public health issue. Social Science & Medicine. 2020;265: 113414. doi:10.1016/j.socscimed.2020.113414

• We added the following reference

Page 13, reference in the following sentence “In the course of November 2020, the main announcement from officials was made from the French President on November 24th, to announce that the vaccination campaign would be “swift” and “massive”, but that getting vaccinated would not be compulsory” was added 

France 24. “We must do everything to avoid a third wave and lockdown,” says Macron. France 24. 24 Nov 2020. Available: https://www.france24.com/en/europe/20201124-live-macron-addresses-the-nation-on-path-out-of-lockdown. Accessed 6 Dec 2021.

Reviewer's Responses to Questions

Reviewer #1: The manuscript is well written, simple, structured language used, and properly organized. Other than some minor problem mentioned in the comments, this research article is quite interesting and easy to understand.

1. In Page 5, Since the cohort procedure is published in another publication, the phrase "Since the cohort protocol is detailed in another publication," requires a bit more background information.

Additional information on the cohort protocol was added in the manuscript, namely on how estimators representative of the French population were produced. 

Underlined elements were added to the cohort description:

Since the cohort protocol is detailed in another publication [11], only the essential characteristics will be presented. A stratified random sample of 135,000 people aged 15 and over, was drawn from the tax database of the National Institute of Statistics and Economic Studies (INSEE), which covers 96% of the population living in France but excludes people living in institutional settings, participated in a first wave of the study in May 2020. People belonging to the lowest decile of income were over-represented. A total of 134,391 respondents participated in the first wave of the study (May 2020). A second wave was conducted in November 2020, including questions on attitudes toward vaccination. Respondents who took part in the first wave of the study were invited to take part in this second wave. In all, 107,808 respondents participated in this second wave (81.7% of the respondents of the first wave of the study). Individuals were invited to answer the questionnaire online, or by phone for those who did not have Internet access. Furthermore, a random sample of 10% of people with Internet access was interviewed by phone in order to take into account a method collection effect. The results published in the study have been adjusted by applying the weights established by the National Institute of Statistics and Economic Studies (INSEE) and marginal recalibration in the survey and sampling design to correct for non-participation, so as to produce estimators that are representative of the population. More information about the cohort can be found in another publication [11].

2. In page 6, It would be better to include the number of people who tested positive for Covid-19 (N=4,036) and whose intention to be vaccinated could be influenced by this fact in the statement "We chose not to include people who tested positive for Covid-19 (N=4,036) and whose intention to be vaccinated could be influenced by this fact."

After thoughtful consideration, we decided not to include the number of people who tested positive for Covid-19 before the survey. Back in November 2020, the message delivered by the French authorities and later confirmed by instructions given to vaccination centers (https://www.rfcrpv.fr/wp-content/uploads/2021/03/N%C2%B0-118-Novembre-2020-f%C3%A9vrier-2021.pdf), was that individuals who had tested positive for Covid-19 a couple of months before, could have to wait a couple of months to be vaccinated. Therefore, as the aim of this article was to study factors associated with reluctance to getting vaccinated, we decided not to include those who had tested positive for Covid-19 in this article, as it is too specific a scenario to be considered with the rest of the population. However, it could be interesting to study, in another article, the attitude of this population on vaccination issues. 

3. On Page 10, insert a square bracket"]" in the statement "The third hypothesis involved a gendered relationship to health and environmental risks, which is also the product of primary socializations [29."

A square bracket was inserted.

4. Modify and correct the phrase structure of the statement "The older people were, the less they refused vaccination" on page 11.

The sentence was replaced with “As they got older, respondents were less likely to refuse vaccination.”

5. In page 13, Instead of giving a direct link, such as "“(https://www.santepubliquefrance.fr/etudes-et-enquetes/coviprev-une-enquete-pour suivre-l-evolution-des-comportements-et-de-la-sante-mentale-pendant-l-epidemie-de covid- 19#block-249162)” it would be better to make it a reference and cite in.

The reference was added and cited in the reference list. 

6. Few references in the text are from 2004 and 2010, and efforts should be made to provide relevant recent references that are no more than five years old.

The reference list was changed according to this comment. References from 2004 and 2010 were removed/replaced by a more current reference.

Reviewer #2: In the current research article entitled "The social specificities of hostility toward vaccination against Covid-19 in France", by Bajos et al., have studied/surveyed opposition toward vaccination against Covid-19 in France. They found that, in France a gendered reluctance toward vaccination in general but even more so regarding vaccination against COVID-19. Further, not only people at the bottom of the social hierarchy but also immigrants and descendants of immigrants were all more reluctant to the Covid-19 vaccine. This article addresses a research topic of great interest, which is under intense investigation in the past 2 years and the manuscript is generally well-written. However, this reviewer has certain suggestions that would help produce a more comprehensive overview of the topic:

Suggestions:

1. At least one additional Figure (illustration) may be provided as to highlight the summary or prospect of this study.

A flowchart of the national EpiCoV cohort was added (Fig 1) following the Sample information section, so as to provide better understanding of the study design. 

2. One paragraph can be added to discussion the Government of France efforts to improve COVID-19 vaccination during the survey period.

The following paragraph has been added in the discussion: 

“It should be added that during the period of the survey, the vaccination campaign was still being developed by French authorities. In the course of November 2020, the main announcement from officials was made from the French President on November 24th, to announce that the vaccination campaign would be “swift” and “massive”, but that getting vaccinated would not be compulsory [40]. It would be coherent to assume that the announcement did not influence the attitude towards the Covid-19 vaccine, and that the link between confidence in the government and reluctance to getting vaccinated was developed before the survey period.”

---

## [Editor Report · Decision Letter 1]

20 Dec 2021

The social specificities of hostility toward vaccination against Covid-19 in France

PONE-D-21-30412R1

Dear Dr. Bajos,

We’re pleased to inform you that your manuscript has been judged scientifically suitable for publication and will be formally accepted for publication once it meets all outstanding technical requirements.

Kind regards,

Sanjay Kumar Singh Patel, Ph.D.

Academic Editor

PLOS ONE
---

## [Editor Report · Acceptance letter]

27 Dec 2021

PONE-D-21-30412R1 

The social specificities of hostility toward vaccination against Covid-19 in France  

Dear Dr. Bajos:

I'm pleased to inform you that your manuscript has been deemed suitable for publication in PLOS ONE. Congratulations! Your manuscript is now with our production department. 

Kind regards, 

on behalf of

Dr. Sanjay Kumar Singh Patel 

Academic Editor

PLOS ONE